# Familial Combined Hyperlipidemia (FCH) Patients with High Triglyceride Levels Present with Worse Lipoprotein Function Than FCH Patients with Isolated Hypercholesterolemia

**DOI:** 10.3390/biomedicines8010006

**Published:** 2020-01-06

**Authors:** Núria Puig, Inka Miñambres, Sonia Benítez, Pedro Gil, Margarida Grau-Agramunt, Andrea Rivas-Urbina, Antonio Pérez, José Luis Sánchez-Quesada

**Affiliations:** 1Cardiovascular Biochemistry Group, Research Institute of the Hospital de la Santa Creu i Sant Pau (IIB Sant Pau), 08041 Barcelona, Spain; npuigg@santpau.cat (N.P.); sbenitez@santpau.cat (S.B.); mgrauag@santpau.cat (M.G.-A.); arivas@santpau.cat (A.R.-U.); 2Biochemistry and Molecular Biology Department, Universitat Autònoma de Barcelona, 08193 Cerdanyola del Valles, Spain; 3Endocrinology and Nutrition Department, Hospital de la Santa Creu i Sant Pau, 08041 Barcelona, Spain; iminambres@santpau.cat (I.M.); pgimll@icloud.com (P.G.); 4CIBER of Diabetes and Metabolic Diseases (CIBERDEM), 28029 Madrid, Spain

**Keywords:** familial combined hyperlipidemia, lipoproteins, lipoprotein function, inflammation, triglycerides

## Abstract

Lipoprotein characteristics were analyzed in familial combined hyperlipidemia (FCH) patients before and after statin treatment. Twenty-six FCH patients were classified according to the presence (HTG group, *n* = 13) or absence (normotriglyceridemic (NTG) group, *n* = 13) of hypertriglyceridemia. Fifteen healthy subjects comprised the control group. Lipid profile, inflammation markers, and qualitative characteristics of lipoproteins were assessed. Both groups of FCH subjects showed high levels of plasma C-reactive protein (CRP), lipoprotein-associated phospholipase A2 (Lp-PLA2) activity and apolipoprotein J. Statins reverted the increased levels of Lp-PLA2 and CRP. Lipoprotein composition alterations detected in FCH subjects were much more frequent in the HTG group, leading to dysfunctional low-density lipoproteins (LDL) and high-density lipoproteins (HDL). In the HTG group, LDL was smaller, more susceptible to oxidation, and contained more electronegative LDL (LDL(-)) compared to the NTG and control groups. Regarding HDL, the HTG group had less Lp-PLA2 activity than the NTG and control groups. HDL from both FCH groups was less anti-inflammatory than HDL from the control group. Statins increased LDL size, decreased LDL(-), and lowered Lp-PLA2 in HDL from HTG. In summary, pro-atherogenic alterations were more frequent and severe in the HTG group. Statins improved some alterations, but many remained unchanged in HTG.

## 1. Introduction

In addition to the quantitative alterations of the lipid profile, it is well-known that the qualitative properties of lipoproteins are a strong determinant of the cardiovascular risk. Thus, characteristics of low-density lipoprotein (LDL), such as particle size, susceptibility to oxidation, aggregability, proportion of electronegative LDL (LDL(-)), or altered composition, have been associated with its atherogenicity [1,2,3]. On the other hand, it has been widely reported that high-density lipoprotein (HDL) exerts atheroprotective functions by counteracting the oxidation of LDL, the inflammatory and apoptotic effects of modified LDL, inhibiting platelet coagulation mediated by an anti-platelet activating factor (PAF) effect, as well as by promoting cholesterol efflux [4,5,6,7]. Several characteristics or components of HDL, such as particle size, lipoprotein-associated phospholipase A_2_ (Lp-PLA_2_) and paraoxonase activity, apolipoprotein (apo) A-I/apoA-II ratio, and apoC-III and apoJ (also known as clusterin) content [5,8,9,10], are related to these antiatherogenic properties, which are compromised in disorders with high risk for coronary artery disease [10,11].

Familial combined hyperlipidemia (FCH) is a common heterogeneous lipid disorder with multigenic origin and a complex pattern of inheritance [12], and it is the most prevalent familial hyperlipemia and a major cause of 20–40% of myocardial infarctions [13]. FCH is associated with high risk for coronary artery disease, which has been related to alterations in lipid profile as well as to the presence of low-grade inflammation [14,15]. This disorder is characterized by a high interindividual and intraindividual variation throughout the patient’s life, and in addition to elevated levels of plasma cholesterol, it can also display with or without increased triglycerides [12,14,16,17]. However, despite its relevance in cardiovascular diseases, FCH is frequently undiagnosed and very often undertreated [17]. The main cause is the heterogeneous phenotypic pattern in which it is displayed; patients have mixed hyperlipemia, isolated hypercholesterolemia, or isolated hypertriglyceridemia. The variability of the phenotypic expression of FCH is relevant to administrating the adequate therapy to each patient at each moment. It is well-known that cardiovascular risk depends not only on blood lipid levels but also on the biological function of lipoproteins. 

Some studies have shown that in FCH, the qualitative properties of lipoproteins are altered, including the chemical composition and characteristics of very low-density lipoprotein (VLDL), LDL, and HDL [12,18]. FCH subjects present a high prevalence of small, dense atherogenic LDL particles (sdLDL) [19,20,21,22], LDL susceptible to oxidation [19,23], and increased levels of circulating oxidized LDL [24,25]. In addition, the size of HDL is smaller in FCH subjects [21]. These qualitative properties differ strongly between subjects with hypercholesterolemia and those with hypertriglyceridemia [26]. However, no study has evaluated the qualitative alterations of lipoproteins depending on the phenotype of FCH patients. Only Georgieva et al. has studied FCH patients classified according to their LDL subclass phenotype, but it focused mainly on the composition of VLDL subclasses [22]. 

Patients with FCH require lifestyle changes and a lipid-lowering treatment that typically consists of potent statins, but it is currently unknown whether statin treatment exerts similar changes on the properties of lipoproteins in the presence and absence of hypertriglyceridemia. The current study determines the qualitative characteristics of VLDL, LDL, and HDL in FCH patients, according to the presence of high plasma triglyceride (TG) levels, and it analyzes the effect of statin therapy.

## 2. Experimental Section

### 2.1. Study Population

Twenty-five FCH patients were included in the study. Criteria for FCH diagnosis were having at least two first-degree relatives with mixed hyperlipidemia or combination of phenotypes, and having LDL-c >160 mg/dL (4.14 mmol/L) and/or triglycerides >200 mg/dL (2.25 mmol/L), according to the Consensus Document of the Spanish Foundation for Hypercholesterolemia [27]. Patients with secondary causes of dyslipidemia, including poorly controlled diabetes mellitus, hypothyroidism, and hepatic or renal impairment, were excluded. Twenty ml of blood in EDTA-containing Vacutainer tubes (Becton Dickinson, Franklin Lakes, NJ, USA) were collected from each participant. Blood was centrifuged at 4 °C for 15 min at 2500 rpm. After determination of the lipid profile, FCH subjects were classified according to their TG levels: Those with fasting TG higher than 2.26 mmol/L were included in the hypertriglyceridemic (HTG) group, and those with fasting TG levels below 2.25 mmol/L were included in the normotriglyceridemic (NTG) group. Following standard clinical practice guidelines, patients with FCH were examined before and after treatment for their dyslipidemia for a minimum of three months. All patients received statin therapy using a starting dose appropriate for the level of LDL-c reduction required to achieve LDL-c <2.6 mmol/L. Appendix A shows the specific statin and dose for each patient. Fifteen healthy subjects comprised the control group. Controls were normolipemic and normoglycemic and had no antecedents of inflammatory, infectious, or coronary disease, or other diseases known to affect lipid metabolism. Clinical and anthropometric data of all groups are shown in Table 1. All subjects gave written informed consent before participating in the study, and the protocol was approved by the ethical committee of the Hospital de la Santa Creu i Sant Pau (IIBSP-REL-2017-27, 26 July 2017). The study was performed in accordance with the Helsinki Declaration.

### 2.2. Plasma Determinations

Lipid profile, C-reactive protein (CRP), total apoJ, Lp-PLA_2_ activity, LDL size, HDL subfractions proportion, adiponectin, and leptin were determined in plasma obtained in EDTA-containing Vacutainer tubes. Lipid profile included total cholesterol, triglycerides, apoB, and VLDL, LDL, and HDL cholesterol. Cholesterol of lipoprotein fractions was quantified using a direct HDL-cholesterol method (HDL-C plus) or by ultracentrifugation when TG concentration was higher than 3 mmol/L, according to the National Cholesterol Education Program [28]. All these determinations and CRP were performed in a Cobas 6000/c501 autoanalyzer using reagents from Roche Diagnostics (Bassel, Switzerland). Adiponectin, leptin (Life Technologies, Carlsbad, CA, USA) and apoJ (Mabtech, Stockholm, Sweden) plasma levels were measured with commercial ELISA kits. LDL size and HDL subfraction proportions were evaluated from serum by nondenaturating polyacrylamide gradient (2.5%–16%) gel electrophoresis (GGE), as described previously [29]. Lp-PLA_2_ activity was determined using 2-tio-PAF (Cayman Chemicals, Ann Arbor, MI, USA) as a substrate [30] according to the manufacturer′s instructions. The distribution of Lp-PLA_2_ between lipoprotein fractions was measured by precipitating apoB-containing lipoproteins from plasma with dextran sulphate [31]. 

### 2.3. Lipoprotein Isolation and Composition

Lipoproteins were isolated by flotation sequential ultracentrifugation according to density: VLDL (1.006–1.019 g/mL), LDL (1.019–1.063 g/mL), and HDL (1.063–1.210 g/mL). Their lipid and apolipoprotein composition was determined by measuring the content of cholesterol, triglycerides, apoB, apoA-I (Roche Diagnostics), phospholipids, free cholesterol (Wako Pure Chemical, Osaka, Japan), apoA-II, apoE, and apoC-III (Kamiya Biomedicals, Seattle, WA, USA) in the autoanalyzer. ApoJ was evaluated using commercial ELISA (Mabtech, Stockholm, Sweden). 

### 2.4. LDL Functional Assays 

LDL susceptibility to oxidation: LDL was dialyzed against phosphate-buffered saline (PBS) pH 7.4 using gel filtration chromatography in a PD10 column (Sephadex G-25, GE Healthcare, Chicago, IL, USA). Susceptibility to oxidation was evaluated by monitoring the formation of conjugated diene formation at 234 nm in a Synergic HT spectrophotometer (BioTek, Winooski, VT, USA). LDL at 0.15 mM of cholesterol was incubated with 5 μM of CuSO_4_, and the lag phase time of the oxidation kinetics was determined [32]. 

Electronegative LDL (LDL(-)): The proportion of LDL(-) was quantified from total LDL using stepwise anion exchange chromatography in a MonoQ 5/50 GL column (GE Healthcare), as described previously [33].

### 2.5. HDL Function Assays

All assays were performed with HDL dialyzed against PBS. 

#### 2.5.1. Antioxidant Capacity of HDL 

HDL at 0.15 mM of cholesterol was incubated with a standard LDL (obtained from a pool of normolipemic plasma and stored with 10% sucrose at −80 °C), and oxidation was induced by adding 5 µM of CuSO_4_. Conjugated diene formation was monitored as explained in the LDL section. Results were expressed as the capacity of HDL to prolong the lag phase time of the standard LDL alone, as described previously [34].

#### 2.5.2. Cholesterol Efflux

Cholesterol efflux induced by HDL was performed using the cell line of macrophages J774A.1 (ATCC TIB-67^TM^) loaded with 3H-cholesterol. Essentially, the assay was conducted as described by Escolà-Gil et al. [35], using as the cholesterol acceptor HDL dialyzed in PBS at 50 mg/L of apoA-I. Data are expressed as the percentage of 3H-cholesterol effluxed from cells by HDL.

#### 2.5.3. Anti-Inflammatory Activity of HDL 

The anti-inflammatory effect of HDL was evaluated in the monocytic line THP1-XBlue^TM^-MD2-CD14 (InvivoGen, San Diego, CA, USA), as described previously [36]. Inflammation was promoted by in vitro oxidized LDL (oxLDL). Monocytes were incubated with HDL (50 µg/mL apoA-I), oxLDL (0.2 mM cholesterol), or oxLDL plus HDL for 24 h. Cell supernatants were collected after 24 h, and IL-6 release was evaluated by ELISA (eBiosciences, Thermo Fisher, Waltham, MA, USA). The effect of oxLDL alone is considered as 100% of inflammation. Data are expressed as the percentage of IL-6 release induced in the presence of HDL.

### 2.6. Statistical Analysis

Statistical analysis was performed using GraphPad Prism 6.0 software. Unpaired data (NTG versus HTG versus control groups) were compared using the nonparametric Mann–Whitney test. Paired data (HFC patients before and after treatment) were analyzed using the nonparametric Wilcoxon test. Correlation analysis was conducted using the Spearman test for nonparametric variables, using the entire studied population (*n* = 40). Data are expressed as mean ± SD or mean ± SEM. *p* < 0.05 was considered significant.

## 3. Results

All tables and figures have the same symbols: * specifically indicates significant changes after treatment in each group; a, b, c, and d mean significant differences vs. basal NTG, NTG post-treatment, basal HTG and HTG post-treatment, respectively.

### 3.1. Lipid Profile and Inflammation Markers

Table 1 shows the anthropometric and clinical characteristics, lipid profile, and inflammatory markers of both groups of FCH patients before and after treatment and of control subjects. As expected, all lipid parameters were higher in patients with FCH, except for HDL-c, which was notably lower in the HTG group. Lipid-lowering therapy decreased total cholesterol and LDL-c, with decrements of 30% and 37%, respectively, without differences between the NTG and HTG groups. TG and VLDL-c decreased in both groups, but still remained higher compared with the control group. ApoB was normalized in both FCH groups. HDL-c was low in the HTG group, and no effect was observed after therapy. CRP concentration was increased in the NTG and HTG groups compared with the control group, and therapy decreased these levels in both groups. No differences were found in adiponectin or leptin between the groups. 

### 3.2. Distribution of Lp-PLA_2_ Activity in Plasma

Lp-PLA_2_ activity was increased in the NTG and HTG groups compared with control subjects (Figure 1a), due mainly to increased activity in apoB-containing lipoproteins but not in HDL (Figure 1b,d). This resulted in a decreased proportion of Lp-PLA_2_ associated with HDL in both groups of FCH patients (Figure 1c). Lipid-lowering therapy normalized the total activity in both FCH groups, due mainly to a decrease in apoB-containing lipoproteins.

### 3.3. Concentration and Distribution of Apolipoprotein J in Plasma

FCH patients presented increased levels of plasma apoJ (Figure 2a), with only a minor part bound to lipoproteins (15%–30%; Figure 2c). The proportion of apoJ associated with lipoproteins was lower in the HTG group compared with controls. This was especially apparent in VLDL (statistical differences in both in NTG and HTG) and LDL (statistical differences only in HTG) (Figure 2d,e). The content of apoJ in HDL was also lower in FCH patients, but statistically significant differences were not achieved (Figure 2f). No significant effect of therapy was observed either in the total plasma apoJ concentration or in its distribution among lipoproteins.

### 3.4. Very Low-Density Lipoprotein (VLDL) Composition 

Figure 3 shows that VLDL from HTG had less cholesterol, phospholipids, and apoB, and more triglycerides and apoC-III compared to VLDL from NTG. In contrast, VLDL from HTG exhibited less cholesterol and apoE and more triglycerides and apoC-III than VLDL from the control group. VLDL from NTG and controls were more similar; the only differences were the presence of less free cholesterol and phospholipids and more apoE and apoC-III in VLDL from the control group. After therapy, cholesterol decreased in both groups of FCH, whereas triglycerides increased only in NTG. 

### 3.5. Low-Density Lipoprotein (LDL) Composition

Figure 4 shows that LDL from HTG contained more protein and apoB and less cholesterol and phospholipids than LDL from NTG and control groups. TG and apoC-III content were also higher in LDL from HTG compared with the control group. The only differences between NTG and controls were increased cholesterol and apoE in the latter. Statin treatment decreased protein and apoB and increased free cholesterol in the HTG group. The only effect of statin treatment on the NTG group was an increase in the apoE content.

### 3.6. LDL Qualitative Properties

Differences in the composition of LDL from HTG patients were suggestive of decreased LDL size, a feature that was confirmed by GGE (Figure 5a). In addition, LDL from the NTG group was also smaller than LDL from control subjects. Therapy increased the LDL size in HTG patients. 

Small LDL size is usually accompanied by increased susceptibility to oxidation. This is what we observed in our samples, although statistically significant differences were observed only when comparing HTG patients with the control group (Figure 5b). No effect of lipid-lowering therapy was observed. The proportion of the atherogenic subfraction LDL(-) was increased in both groups of FCH patients compared with the control group (Figure 5c). Therapy slightly decreased this proportion in the HTG group, but it still remained far from control values. 

### 3.7. High-Density Lipoprotein (HDL) Composition 

Compared with HDL from control subjects, HDL from HTG patients had lower apoA-I, cholesterol and phospholipid content and higher TG, apoA-II and apoC-III content (Figure 6). HDL from HTG contained also more triglycerides than HDL from NTG. The only differences between NTG and controls were increased apoA-II and apoC-III. Considering the ratio apoA-I/apoA-II, both groups of FCH presented a lower ratio than control subjects. Therapy increased apoA-I in NTG and decreased apoE and apoC-III in both groups of FCH.

### 3.8. HDL Qualitative Properties

As a whole, the composition data of HDL indicate that HDL from HTG patients should present a lower proportion of large, mature HDL2 particles than HDL from control subjects. This was confirmed by GGE, which also showed a similar pattern in HDL from the NTG group (Figure 7a). Therapy slightly increased the proportion of HDL2, but without observing statistical significance. The capacity to promote cholesterol efflux or the antioxidant potential to inhibit LDL oxidation was similar in all the groups and was unaffected by therapy (Figure 7b,c). However, the anti-inflammatory effect of HDL from FCH patients was lower than that of HDL from the control group, finding statistically significant differences in the NTG group (Figure 7d). No effect of lipid-lowering treatment was observed. 

### 3.9. Correlation of Triglyceride Plasma Levels with the Qualitative Properties of Lipoproteins

To gain insight into the association between high TG levels in plasma and the rest of the measured variables, a Spearman correlation test was conducted. These data are shown in Appendix A. As expected, plasma TG levels correlated with all the parameters of the lipid profile. Regarding inflammation-related parameters, triglycerides correlated positively with CRP, total Lp-PLA_2_ activity, and Lp-PLA_2_ associated with apoB-containing lipoproteins, but not with leptin or adiponectin. Regarding the association of plasma triglycerides with lipoprotein composition, they were surprisingly poorly related with VLDL components (correlated negatively only with apoB and apoJ and positively only with apoC-III), but they were strongly related with all LDL components except apoE. Accordingly, plasma triglycerides were negatively correlated with LDL size and susceptibility to oxidation and positively with the proportion of LDL(-). Finally, regarding HDL characteristics, plasma triglycerides were negatively correlated with cholesterol, phospholipids, apoA-I/apoA-II ratio, and HDL2 proportion and were associated positively with triglycerides and apoC-III. 

## 4. Discussion

The present study shows that statin therapy efficiently normalizes lipid parameters related mainly to cholesterol metabolism (total cholesterol, LDL-c) but that the parameters related mainly to TG metabolism (VLDL-c, HDL-c) remain abnormal after therapy in subjects with FCH. The study’s main and novel finding is that the persistence of lipoprotein disorders following statin therapy is much more marked in patients with HTG. These alterations could compromise lipoprotein functionality and contribute to the especially high residual risk in this subgroup of patients under treatment with statins only. These findings support current recommendations aimed at treating hypertriglyceridemia in patients with stable statin therapy, to achieve additional risk reduction [37].

FCH is associated with high risk for coronary artery disease related to lipoprotein abnormalities. It is well-known that cardiovascular risk depends not only on blood lipid levels but also on the biological function of lipoproteins. These qualitative properties differ strongly between subjects with hypercholesterolemia and those with hypertriglyceridemia [24]. However, no study has evaluated the qualitative alterations of lipoproteins depending on the phenotype of FCH patients and the effect of statin therapy. Our data show that some lipoprotein alterations persist after therapy in both groups of FCH patients but that this was more evident in HTG subjects.

Regarding inflammation-related parameters, we found that the high levels of CRP observed in FCH decrease after lipid-lowering therapy. In addition, total and apoB-associated Lp-PLA_2_ activity showed the same pattern as CRP, with high levels in both groups of FCH patients and a significant decrease after therapy. In contrast, the adipokines leptin and adiponectin exhibited similar plasma levels between control subjects and FCH patients. These observations suggest that systemic inflammation in FCH may not be related with adiposity (BMI was similar in the three groups) but rather may be due to alterations in the lipid metabolism and perhaps due to the presence of underlying atherosclerotic disease.

Another important observation is that the plasma concentration of apoJ was increased in both FCH groups. Paradoxically, the proportion of apoJ bound to lipoproteins was lower in the HTG group, due mainly to a low proportion associated with VLDL and LDL, which concurs with a previous study by our group [38]. The content of apoJ in lipoproteins is altered in patients with coronary heart disease or with cardiovascular risk factors [9,39,40]. The physiological meaning of this finding is unclear, but several studies have reported that apoJ plays a protective role when it is associated with lipoproteins through different mechanisms. ApoJ inhibits the cytotoxic properties of modified LDL [41], and poor apoJ content in HDL yields particles with poor antiapoptotic activity. In addition, apoJ prevents LDL aggregation from acting as a chaperone that stabilizes the structure of apoB [42]. Therefore, the low apoJ content in lipoproteins from both HFC groups, but especially in HTG subjects, may compromise their functionality.

The alterations in the composition of VLDL from the HTG group compared with control subjects are suggestive of increased size (more triglycerides and less apoB and cholesterol content). This observation is in agreement with data from Georgieva et al. [22], who grouped FCH patients according to the phenotype of LDL subclasses. Although these authors used an approach slightly different from ours, they also separated patients by TG concentration. The larger size of VLDL in HTG may be related to our observation of high apoC-III content in these patients, because this apolipoprotein is a central regulator of VLDL catabolism acting as a specific inhibitor of lipoprotein lipase, that is, retarding the catabolism of VLDL particles [43]. On the other hand, apoE content was low in both FCH groups. Because apoE is the main ligand of lipoprotein receptors that mediate VLDL clearance [44], this would also contribute to poor catabolism of VLDL in plasma, prolonging its mean lifetime in blood. Our data showed that both apoC-III and apoE content in VLDL was unchanged after lipid-lowering treatment. Therefore, the catabolism of VLDL would remain impaired after treatment due to the combined effects of low apoE and high apoC-III content. This observation may partly explain the relatively poor response of triglycerides and VLDL-c to statin therapy.

The consequence of VLDL-impaired catabolism explains the altered composition observed in LDL particles from HTG patients (more triglycerides, apoC-III, and apoB and less cholesterol and phospholipids). As a result, small, dense LDL particles are formed, and these particles are characterized by increased susceptibility to oxidation and a high proportion of modified electronegative LDL particles (LDL(-)) [1,29]. As a whole, these characteristics are highly proatherogenic and proinflammatory [45,46,47,48], and they indicate that LDL from FCH patients with HTG is more pernicious than LDL from controls or FCH patients with NTG. The high inflammatory potential of modified LDLs (oxidized, electronegative particles) may contribute to the low-grade inflammation found in FCH patients. In this case, although lipid-lowering therapy normalized LDL-c values in plasma, only a slight improvement in LDL size and the proportion of LDL(-) was observed in HTG patients, without reaching the values of control subjects.

Regarding HDL particles, our data show that HDL from HTG patients has more triglycerides and less apoA-I, cholesterol, and phospholipids than HDL from controls. These properties are compatible with small particle size, and we consequently observed a lower proportion of large HDL2 particles in FCH than in controls. The ratio apoA-I/apoA-II is also lower, whereas apoC-III content is higher in both groups of FCH compared with the control group. The role of apoC-III in HDL is unclear, but it is generally associated with dysfunctional particles and increased cardiovascular risk [49,50]. An increased content of apoC-III in HDL has been related to the loss of antiapoptotic capacity in HDL [9]. In this context, we found that the ability of HDL to prevent the release of IL-6 induced by oxidized LDL in monocytes is diminished in HDL from FCH patients. Although we have no direct evidence of the role of apoC-III in the anti-inflammatory ability of HDL, in vitro and in vivo studies have shown that this apolipoprotein can stimulate the expression of inflammation mediators in monocytes and muscle cells and that overexpression in mice increases inflammation in the aorta [51,52,53].

The correlation analysis between plasma triglycerides and the rest of the parameters provided results that are in good agreement with the comparisons between groups. Besides the expected correlations (i.e., lipid profile or inflammation parameters), the most interesting finding was that although most triglycerides are transported in VLDL, the levels of plasma triglycerides are much more strongly related to the composition of LDL and HDL than to that of VLDL. This is because the catabolism of VLDL determines the composition and biological characteristics of LDL and HDL. Thus, it becomes clear that plasma triglycerides are extremely important for appraising cardiovascular risk, not specifically because of their levels but because they modulate the atherogenic or antiatherogenic properties of LDL and HDL.

The main limitation of the study relates to the low number of subjects analyzed, which increases the likelihood of nonsignificant findings as a result of low statistical power. Nevertheless, despite this power limitation, the alterations in the composition and function of the different lipoproteins are coherent and metabolically connected, and the correlation analysis between plasma triglycerides and the rest of the evaluated parameters confirms most of the differences observed between the groups and supports the validity of our results. A second limitation is the time of therapy since it is possible that a prolonged treatment could have further effects on lipoprotein function that would have not been detected in the present study. Finally, another limitation of our work is that, as an observational study, it does not provide insight into the specific molecular mechanisms by which alterations in the composition of lipoproteins affect the development of atherosclerosis.

## 5. Conclusions

In summary, our data show numerous abnormalities in the lipoproteins of both groups of FCH patients, but these are much more abundant in the HTG group. Besides lipid profile and inflammatory markers, the current study has analyzed 47 different parameters in VLDL, LDL, and HDL. A summary of all detected differences compared with control subjects yields 13 statistically significant differences in the NTG group versus 32 in the HTG group. Moreover, although statin treatment improved some characteristics of the lipoproteins, many of the alterations detected in the HTG group remained unchanged. These observations support the notion that HFC patients displaying HTG are at higher cardiovascular risk than those with only high cholesterol levels. Hence, more aggressive lipid-lowering therapies, including approaches targeting hypertriglyceridemia, should be considered when a phenotype of high TG levels is present in these patients.

## Figures and Tables

**Figure 1 biomedicines-08-00006-f001:**
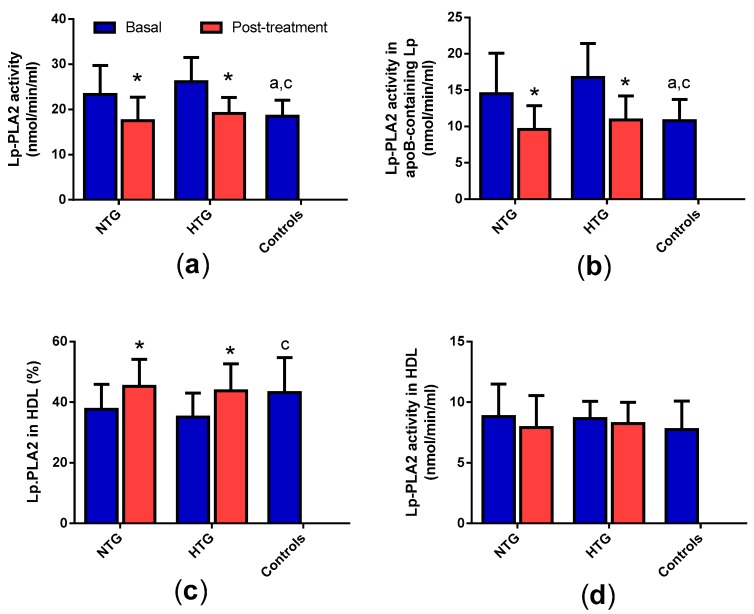
Total Lp-PLA_2_ activity and distribution in lipoproteins. (**a**) Total activity; (**b**) and (**d**): Activity associated to apoB-lipoproteins and high-density lipoprotein (HDL), respectively; (**c**) proportion of activity associated to HDL. Data are expressed as mean ± SD. * *p* < 0.05 vs. basal samples. ^a^
*p* < 0.05 vs. basal NTG. ^b^
*p* < 0.05 vs. NTG post-treatment. ^c^
*p* < 0.05 vs. basal HTG. ^d^
*p* < 0.05 vs. HTG post-treatment.

**Figure 2 biomedicines-08-00006-f002:**
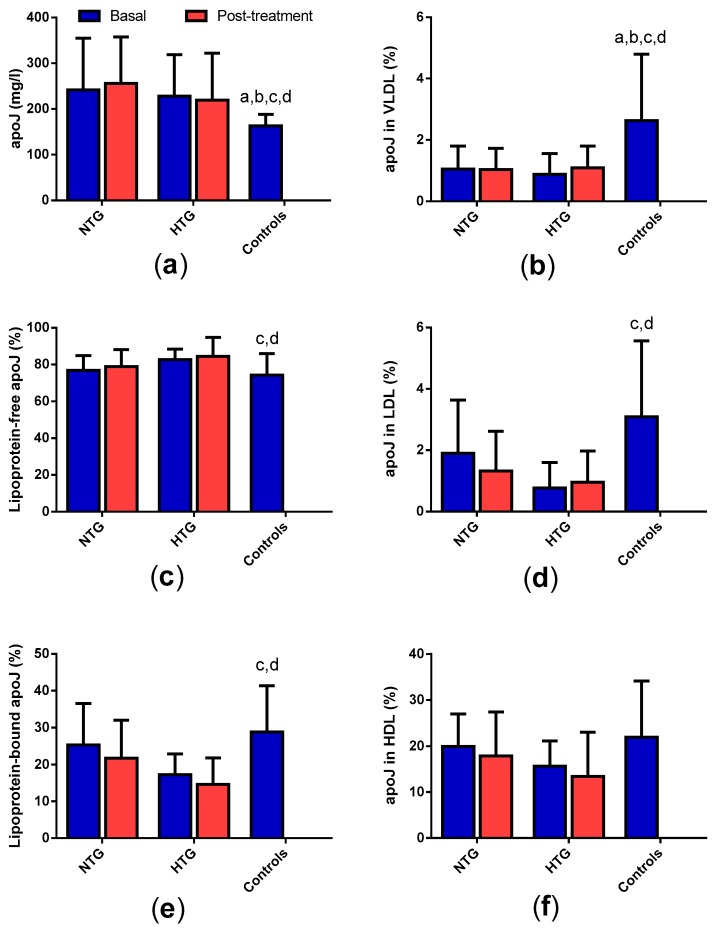
Concentration and distribution of apolipoprotein J (apoJ) in plasma and lipoproteins. (**a**) Total apoJ concentration; (**b**,**d**,**f**) proportion of apoJ associated to very low-density lipoprotein (VLDL), low-density lipoprotein (LDL), and HDL, respectively; (**c**) proportion of apoJ not associated to lipoproteins; (**e**) proportion of apoJ associated to lipoproteins. Data are expressed as mean ± SD. * *p* < 0.05 vs. basal samples. ^a^
*p* < 0.05 vs. basal NTG. ^b^
*p* < 0.05 vs. NTG post-treatment. ^c^
*p* < 0.05 vs. basal HTG. ^d^
*p* < 0.05 vs. HTG post-treatment.

**Figure 3 biomedicines-08-00006-f003:**
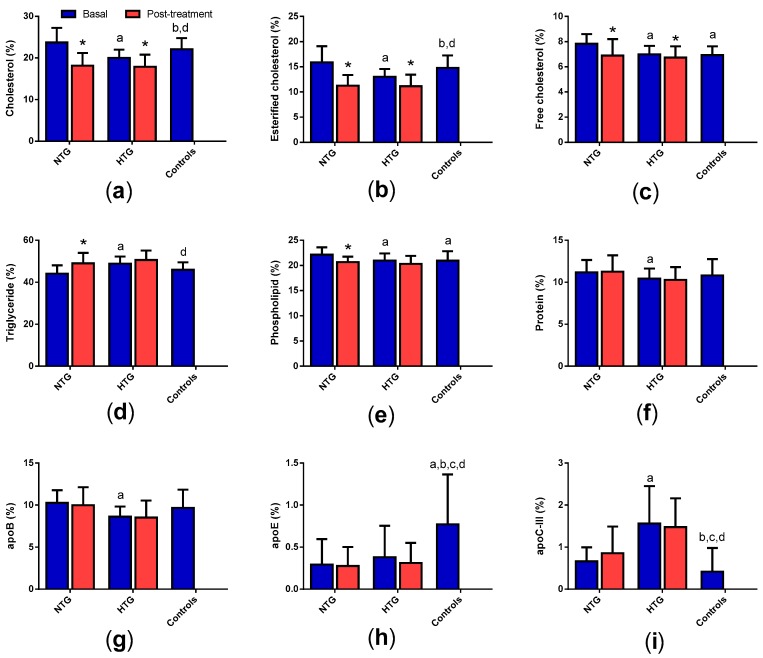
VLDL composition. Data indicate the relative proportion of each component of the VLDL mass and are expressed as mean ± SD. (**a**) total cholesterol, (**b**) esterified cholesterol, (**c**) free cholesterol, (**d**) triglycerides, (**e**) phospholipids, (**f**) total protein, (**g**) apoB, (**h**) apoE, (**i**) apoC-III. * *p* < 0.05 vs. basal samples. ^a^
*p* < 0.05 vs. basal NTG. ^b^
*p* < 0.05 vs. NTG post-treatment. ^c^
*p* < 0.05 vs. basal HTG. ^d^
*p* < 0.05 vs. HTG post-treatment.

**Figure 4 biomedicines-08-00006-f004:**
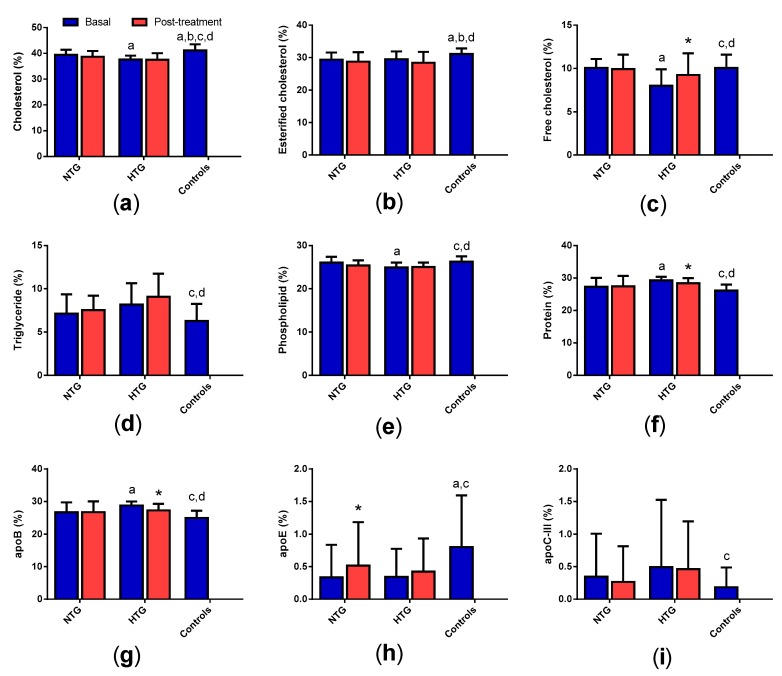
LDL composition. Data indicate the relative proportion of each component of the LDL mass and are expressed as mean ± SD. (**a**) total cholesterol, (**b**) esterified cholesterol, (**c**) free cholesterol, (**d**) triglycerides, (**e**) phospholipids, (**f**) total protein, (**g**) apoB, (**h**) apoE, (**i**) apoC-III. * *p* < 0.05 vs. basal samples. ^a^
*p* < 0.05 vs. basal NTG. ^b^
*p* < 0.05 vs. NTG post-treatment. ^c^
*p* < 0.05 vs. basal HTG. ^d^
*p* < 0.05 vs. HTG post-treatment.

**Figure 5 biomedicines-08-00006-f005:**
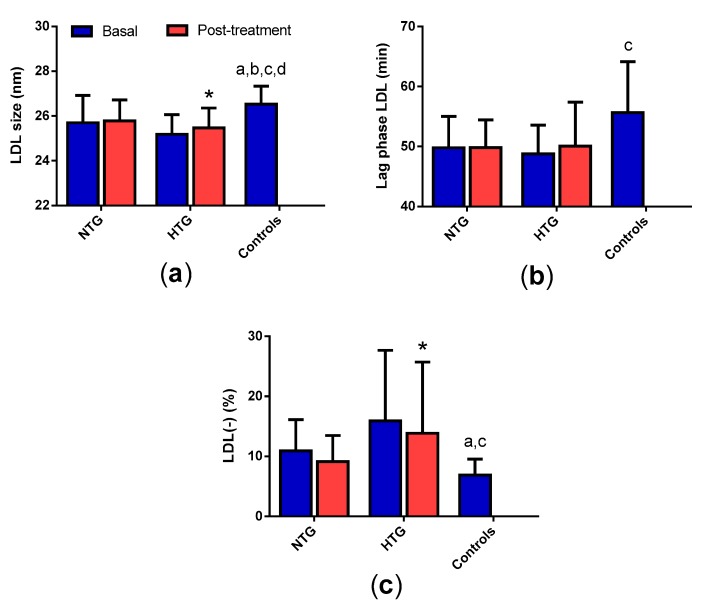
Qualitative properties of LDL. (**a**) LDL particle size was determined by gradient gel electrophoresis (GGE) and expressed as nm. (**b**) Lag phase time of the oxidation kinetics of LDL induced by CuSO_4_ monitoring the formation of conjugated dienes, expressed as min. (**c**) Proportion of electronegative LDL. Data are expressed as mean ± SD. * *p* < 0.05 vs. basal samples. ^a^
*p* < 0.05 vs. basal NTG. ^b^
*p* < 0.05 vs. NTG post-treatment. ^c^
*p* < 0.05 vs. basal HTG. ^d^
*p* < 0.05 vs. HTG post-treatment.

**Figure 6 biomedicines-08-00006-f006:**
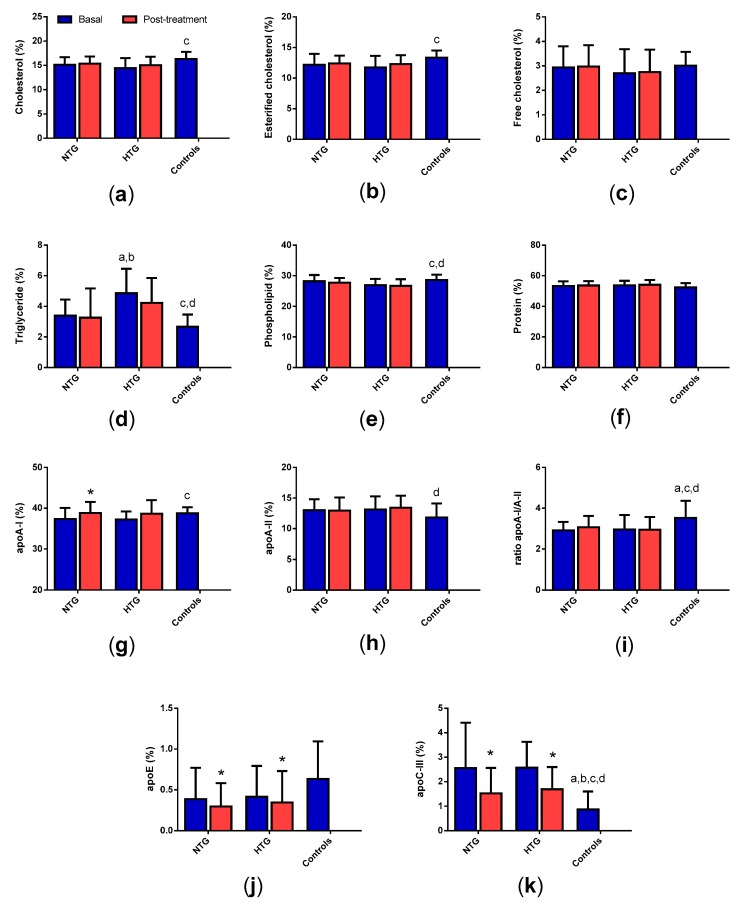
HDL composition. Data indicate the relative proportion of each component of the HDL mass, except the ratio apoA-I/apoA-II, and are expressed as mean ± SD. (**a**) total cholesterol, (**b**) esterified cholesterol, (**c**) free cholesterol, (**d**) triglycerides, (**e**) phospholipids, (**f**) total protein, (**g**) apoA-I, (**h**) apoA-II, (**i**) ratio apoA-I/apoA-II, (**j**) apoE, (**k**) apoC-III. * *p* < 0.05 vs. basal samples. ^a^
*p* < 0.05 vs. basal NTG. ^b^
*p* < 0.05 vs. NTG post-treatment. ^c^
*p* < 0.05 vs. basal HTG. ^d^
*p* < 0.05 vs. HTG post-treatment.

**Figure 7 biomedicines-08-00006-f007:**
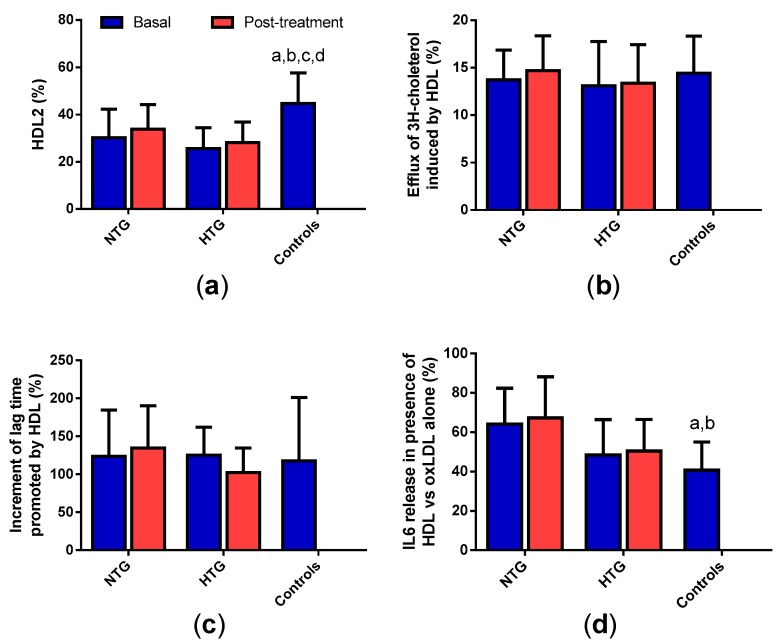
Qualitative properties of HDL. (**a**) HDL2 (large HDL particles) proportion was determined by GGE and expressed as the proportion of total HDL; (**b**) proportion of ^3^H-cholesterol effluxed by HDL from cholesterol-loaded macrophages; (**c**) increment of the lag phase time of the oxidation kinetics of a standard LDL induced by CuSO_4_ in the presence of the HDL from each patient, expressed as percentage of lag time increase. (**d**) Proportion of IL-6 released from monocytes by oxidized LDL (oxLDL) in presence of HDL, compared with monocytes incubated with oxLDL alone (100% of IL-6 release). Data are expressed as mean ± SD. * *p* < 0.05 vs. basal samples. ^a^
*p* < 0.05 vs. basal NTG. ^b^
*p* < 0.05 vs. NTG post-treatment. ^c^
*p* < 0.05 vs. basal HTG. ^d^
*p* < 0.05 vs. HTG post-treatment.

**Table 1 biomedicines-08-00006-t001:** Anthropometric and clinical characteristics, lipid profile, and inflammatory markers.

	NTG Group (*n* = 13)		HTG Group (*n* = 12)		Control Group (*n* = 15)
**Age (years)**	48.6 ± 10.6		51.1 ± 6.6		48.4 ± 6.0
**Gender (M/F)**	7/5		8/4		9/6
	**Baseline**	**After Therapy**	**Baseline**	**After Therapy**	
**BMI (kg/m^2^)**	27.1 ± 4.3	26.5 ± 4.4	27.1 ± 4.7	27.3 ± 4.6	24.7 ± 4.3
**Cholesterol (mmol/L)**	7.18 ± 0.89	4.71 ± 0.66 *	7.44 ± 0.81	5.04 ± 0.91 *	5.44 ± 0.99 ^a,c^
**Triglycerides (mmol/L)**	1.84 ± 0.44	1.40 ± 0.54 *	3.61 ± 0.79 ^a^	2.35 ± 0.73 *^,a,b^	0.93 ± 0.28 ^a,c,d^
**VLDL-c (mmol/L)**	0.87 ± 0.20	0.59 ± 0.20 *	1.51 ± 0.47 ^a^	1.02 ± 0.25 *^,a,b^	0.43 ± 0.13 ^a,c,d^
**LDL-c (mmol/L)**	5.02 ± 0.78	2.73 ± 0.54 *	4.95 ± 0.75	3.03 ± 0.89 *	3.31 ± 0.96 ^a,c^
**HDL-c (mmol/L)**	1.29 ± 0.37	1.28 ± 0.35	0.98 ± 0.14 ^a^	0.99 ± 0.22 ^a,b^	1.51 ± 0.19 ^c,d^
**ApoB (mmol/L)**	1.50 ± 0.19	0.90 ± 0.16 *	1.64 ± 0.19	1.11 ± 0.22 *	0.95 ± 0.18 ^a,c^
**CRP (mg/L)**	2.79 ± 2.45	1.72 ± 1.64 *	2.73 ± 2.51	1.56 ± 1.30 *	1.02 ± 0.63 ^a,c^
**Leptin (µg/L)**	36.3 ± 30.3	43.9 ± 32.3	28.6 ± 33.7	29.5 ± 30,0	33.6 ± 29.7
**Adiponectin (µg/L)**	10.9 ± 2.5	11.1 ± 2.7	10.2 ± 2.1	10.3 ± 1.7	10.2 ± 2.4

* *p* < 0.05 vs. basal samples. ^a^
*p* < 0.05 vs. basal NTG. ^b^
*p* < 0.05 vs. NTG post-treatment. ^c^
*p* < 0.05 vs. basal HTG. ^d^
*p* < 0.05 vs. HTG post-treatment.

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
