# Peer review of "Familial Combined Hyperlipidemia (FCH) Patients with High Triglyceride Levels Present with Worse Lipoprotein Function Than FCH Patients with Isolated Hypercholesterolemia"

_biomedicines, 2020, doi:10.3390/biomedicines8010006_

Round 1

Reviewer 1 Report

Reviewer Evaluation:

This research paper assesses the interesting topic of the functionality of lipoproteins in FCH patient subtypes. Overall the article is well-written and very interesting despite the low statistical power. Overall,  a little more detail in the methodology is required as outlined below.

General Comments:

Line 43 – HDL also exhibits an antiplatelet effect and anti-PAF effect than can be cardioprotective - the following discuss functionality of HDL (https://www.ncbi.nlm.nih.gov/pubmed/24891399; doi:10.3390/nu10050604). Other than that, the introduction is well written and to the point. Section 2.2 roche (provide details of the place of manufacture etc. Maybe just state Roche once and not repeat it) How much blood was taken from participants, how were the vacutainers handled etc. Just state as per manufacturers guidelines if that is the case. Otherwise provide details. Section 2.5 with one sentence doesn’t seem to fit. Is this line necessary or do you need to provide more detail for the HDL functional assays. Readers need to be able to replicate the experiments. How many times were fasting bloods taken in the screening process of this study to determine what groups each of the participants belonged to? Table 1 – place ‘Baseline’ and ‘After therapy’ over the cholesterol levels. I’m not sure if a change in BMI was intended, but in this case I’m not sure if before and after is necessary as it was more of a screening tool. Were the measurements for the controls taken at the start and end of the time as well? Or was just one measurement taken, if so when? Title: This is totally optional, but I would replace title as follows to make it simpler:

Familial combined hyperlipidemia (FCH) patients with high triglyceride levels present with worse lipoprotein function than FCH patients with isolated hypercholesterolemia

Minor Points:

Page 1 - Line 25 – the Lp-PLA2 the 2 is usually a subscript, similarly phospholipase A2 is usually subscript e.g. Lp-PLA2 & phospholipase A2

Line 30 – ‘content’ you mean concentration?

Line 30 – from the control

Line 62-3 – this line is repetition

Line 102 – brand of vacutainers?

Line 147 – usually written as IL-6 needs a ‘-‘

Line 150 – developer/manufacturer details

Table 1 – fix the brackets after adiponectin and leptin

Line 226/253 – not achieved… replace with observed… it sounds like you are seeking it, which isn’t the intended manner of statistical testing.

Concluding Remarks:

Overall, the study is very interesting and worth further investigation. Many researchers are beginning to realise the overall functionality of lipoproteins may be more important than the levels. There are weaknesses in the study due to power etc., however, the authors have acknowledged this in the discussion.

Author Response

Replies to reviewer 1

We are grateful for the comments of the reviewer, which will improve the quality of the manuscript.

Line 43 – HDL also exhibits an antiplatelet effect and anti-PAF effect than can be cardioprotective - the following discuss functionality of HDL (https://www.ncbi.nlm.nih.gov/pubmed/24891399; doi:10.3390/nu10050604).

According to the reviewer’ suggestion this interesting review, as well as the anti-aggreging effect of HDL, have been included in the manuscript.

Other than that, the introduction is well written and to the point.

Thanks for the kind comment.

Section 2.2 roche (provide details of the place of manufacture etc. Maybe just state Roche once and not repeat it)

This section has been modified according to the reviewer’ requirements. The place of manufacturer of all reagents has been included.

How much blood was taken from participants, how were the vacutainers handled etc. Just state as per manufacturers guidelines if that is the case. Otherwise provide details.

Twenty ml of blood in EDTA-containing Vacutainer tubes were collected from each participant. Blood was centrifuged at 4ºC for 15 minutes at 2500 rpm. Vacutainer tubes were from Becton Dickinson. This information has been included in the current version of the manuscript.

Section 2.5 with one sentence doesn’t seem to fit. Is this line necessary or do you need to provide more detail for the HDL functional assays. Readers need to be able to replicate the experiments.

The dialysis of HDL in PBS was common to all the subsequent procedures of HDL function, then, this sentence is stated here to avoid unnecessary repetitions in 2.5.1, 2.5.2 and 2.5.3. The methods of lipoprotein function (2.4 and 2.5) are explained in detail in the references 31 to 35.

How many times were fasting bloods taken in the screening process of this study to determine what groups each of the participants belonged to?

Only one blood extraction was performed. The assignment to each group was made after the determination of the lipid profile

Table 1 – place ‘Baseline’ and ‘After therapy’ over the cholesterol levels.

The Table has been modified according to the reviewer’ suggestion

I’m not sure if a change in BMI was intended, but in this case I’m not sure if before and after is necessary as it was more of a screening tool. Were the measurements for the controls taken at the start and end of the time as well? Or was just one measurement taken, if so when?

The main aim of therapy was not a change in BMI, but a change in lipid levels and lipoprotein function. However, we consider of value to include BMI before and after therapy to show that therapy had no effect on BMI. BMI of controls was taken at the time of blood extraction.

Title: This is totally optional, but I would replace title as follows to make it simpler: Familial combined hyperlipidemia (FCH) patients with high triglyceride levels present with worse lipoprotein function than FCH patients with isolated hypercholesterolemia

The title has been modified according to the kind reviewer’ suggestion

Minor Points:

Page 1 - Line 25 – the Lp-PLA2 the 2 is usually a subscript, similarly phospholipase A2 is usually subscript e.g. Lp-PLA& phospholipase A2

Changed according to the reviewer’ suggestion

Line 30 – ‘content’ you mean concentration?

We measured activity of Lp-PLA2, not mass, accordingly, we have changed “content” by “activity”

Line 30 – from the control

Changed according to the reviewer’ suggestion

Line 62-3 – this line is repetition

This sentence has been modified to avoid repetition

Line 102 – brand of vacutainers?

Becton Dickinson, this has been included in the revised version

Line 147 – usually written as IL-6 needs a ‘-‘

Changed according to the reviewer’ suggestion

Line 150 – developer/manufacturer details

Changed according to the reviewer’ suggestion

Table 1 – fix the brackets after adiponectin and leptin

Changed according to the reviewer’ suggestion

Line 226/253 – not achieved… replace with observed… it sounds like you are seeking it, which isn’t the intended manner of statistical testing.

Changed according to the reviewer’ suggestion

Reviewer 2 Report

The authors of the paper “Familial combined hyperlipidemia (FCH) patients displaying high triglyceride levels present worse lipoprotein function than FCH patients with isolated hypercholesterolemia” show that statins therapy efficiently normalizes lipid parameters related mainly to cholesterol metabolism (total cholesterol, LDL-c) but that the parameters related mainly to TG metabolism (VLDLc, HDL-c) remain abnormal after therapy in subjects with FCH. Moreover they point that this study has some limits. This limits make some of the findings non significant.

The authors also state that they do not provide a specific molecular mechanism. In spite of the previous statements the authors show numerous abnormalities in the lipoproteins of both groups of FCH patients and statins treatment improved some characteristics of lipoproteins.

The author should answer several questions.

In all captions the statistics symbols do not clearly match the figures they refer to. The lines 184 and 185 are not clear. In line 186 the authors state that the proportion of ApoJ associated with lipoproteins is lower in the HTG group compared with controls. In figure 2 e however the NTG group also shows lower levels than control. This happens again in figure 2d that does not fully correspond with the text in line 187. In figure 2f, the authors say that shown results are not significant, but what is the reason for this? A low number of patients? Authors should clarify this Figure 3 and text in paragraph between lines 198 and 203 do not match. For instance, they say that there are less phospholipids in HTG than in controls but this in not shown in figure 3e. This happens again in paragraph between lines 210 and 215, the text does not match with figure 4, and for instance figure 4a does not show an increase in cholesterol levels compared to controls. In figure 4i the error bars are too big In lines 229 and 230 the effects of therapy are not shown in figure 5 regarding figure 6 and lines 238 to 243 the same happens again the differences are not clearly shown in the figure. In lines 256-257 the anti-inflamatory effect of HDL from FCH patients is lower than controls but in figure 7d the bars shown it is higher please clarify this. If authors do not find significant differences in statins treatments couldn´t this be due to the short time the patients are studied? All samples are collected after 3 months time of treatment have the authors performed any study after a year time of treatment? The author should review the discussion taking into account all the previous comments

Author Response

Replies to Reviewer 2

The authors of the paper “Familial combined hyperlipidemia (FCH) patients displaying high triglyceride levels present worse lipoprotein function than FCH patients with isolated hypercholesterolemia” show that statins therapy efficiently normalizes lipid parameters related mainly to cholesterol metabolism (total cholesterol, LDL-c) but that the parameters related mainly to TG metabolism (VLDLc, HDL-c) remain abnormal after therapy in subjects with FCH. Moreover they point that this study has some limits. This limits make some of the findings non significant.

The authors also state that they do not provide a specific molecular mechanism. In spite of the previous statements the authors show numerous abnormalities in the lipoproteins of both groups of FCH patients and statins treatment improved some characteristics of lipoproteins.

The author should answer several questions.

We are grateful to the reviewer’ criticisms, that surely can help to improve the clarity of the manuscript. We think that most of the reviewer’ queries are due to the lack of clarity in the symbols indicating statistical differences in the graphs. We hope that changes included in the manuscript will result in a better comprehension of the results.

In all captions the statistics symbols do not clearly match the figures they refer to. The lines 184 and 185 are not clear.

We apologize for the confusion in the statistics symbols. For a better comprehension the term “pre-treatment” in the figure legend has been changed to “basal”, as is indicated in the figure. Accordingly, this change has also been done in all the figure legends. All tables and figures have the same symbols: * specifically indicates significant changes after treatment in each group; whereas a, b, c, and d mean differences between the different groups vs the first, second, third and fourth columns, respectively, in each graph. To improve clarity, we have included a sentence at the beginning of the Results section explaining the meaning of symbols.

In line 186 the authors state that the proportion of ApoJ associated with lipoproteins is lower in the HTG group compared with controls. In figure 2 e however the NTG group also shows lower levels than control. This happens again in figure 2d that does not fully correspond with the text in line 187.

We suppose that the reviewer refers to the whole paragraph 3.3 (the lines do not match with the version of the manuscript I have downloaded from Biomedicines). The reviewer is right, apoJ associated to lipoproteins and to LDL are lower in NTG group compared with controls; however, statistically significant differences were only achieved in the HTG group. The text has been slightly modified for a better comprehension.

In figure 2f, the authors say that shown results are not significant, but what is the reason for this? A low number of patients?

As the reviewer suggests, perhaps increasing the number of patients statistical differences could be found in the apoJ content of HDL. In addition, the dispersion of these data also should account for the lack of significant differences among groups. The low number of studied subjects was included as a limitation of the study.

Authors should clarify this Figure 3 and text in paragraph between lines 198 and 203 do not match. For instance, they say that there are less phospholipids in HTG than in controls but this in not shown in figure 3e.

In the first sentence of the paragraph 3.4 we state that “VLDL from HTG had less cholesterol, phospholipids, and apoB and more triglycerides and apoC-III compared to VLDL from NTG”. But in the second sentence, in which differences between HTG and controls are explained, we say that “VLDL from HTG also exhibited less cholesterol and apoE and more triglycerides and apoC-III than VLDL from the control group”; here we do not indicate any difference in phospholipid content. However, we recognize that perhaps this paragraph is confusing, and the text has been slightly modified to improve clarity.

This happens again in paragraph between lines 210 and 215, the text does not match with figure 4, and for instance figure 4a does not show an increase in cholesterol levels compared to controls. In figure 4i the error bars are too big

In the paragraph 3.5 we state that “LDL from HTG contained more protein and apoB and less cholesterol and phospholipids than LDL from NTG and control groups”. And in the third sentence we indicate that controls have more cholesterol than the NTG group.

Regarding the error bars in Figure 4i, this is due to the very low content of apoC-III in LDL, (the same occurs with apoE), which was close of the detection limit in some samples.

In lines 229 and 230 the effects of therapy are not shown in figure 5

The effect of therapy is indicated with an asterisk (last sentence in the first paragraph of 3.6 “Therapy increased the LDL size in HTG patients”; last sentence in the second paragraph of 3.6 “Therapy slightly decreased this proportion in the HTG group,”), as can be observed in 5a (LDL size) and in 5c (LDL(-)).

regarding figure 6 and lines 238 to 243 the same happens again the differences are not clearly shown in the figure.

As with the other figures, the effect of therapy is indicated with asterisks. This is indicated in the last sentence of the paragraph 3.7: “Therapy increased apoA-I in NTG and decreased apoE and apoC-III in both groups of FCH”

In lines 256-257 the anti-inflamatory effect of HDL from FCH patients is lower than controls but in figure 7d the bars shown it is higher please clarify this.

We apologize for the lack of clarity of this graph. The bars in this figure indicate the percentage of IL-6 released in the presence of HDL versus oxLDL alone, which is the 100% of inflammation. Therefore, the lower the proportion of released IL6 the higher the anti-inflammatory effect of HDL. Accordingly, both the method and the figure legend have been modified to improve clarity. 

If authors do not find significant differences in statins treatments couldn´t this be due to the short time the patients are studied? All samples are collected after 3 months time of treatment have the authors performed any study after a year time of treatment?

The effect of statins on lipid profile occurs rapidly after 2-4 weeks of treatment, and changes in other parameters of lipoprotein function use to occur within 3 months (for instance see Sánchez-Quesada at al Am J Cardiol 1999;84(6):655-9 or Benitez et al Am J Cardiol. 2004;93(4):414-20). However, the reviewer is right and perhaps a prolonged treatment could have further effects on lipoprotein function. Unfortunately, we did not collect blood after one year of therapy. This has been commented in the discussion section as a limitation of the study.

The author should review the discussion taking into account all the previous comments

We are grateful to the reviewer’ criticisms, which undoubtedly will help to improve the clarity of the manuscript. We have included some comments on the possibility that increasing the time of statin therapy could have further protective effect on lipoprotein function. We hope that changes result in a better comprehension of the manuscript.